# COVID-19 Vaccination Hesitancy in Autoimmune Disease Patients: Policy Action and Ethical Considerations

**DOI:** 10.3390/vaccines11081283

**Published:** 2023-07-26

**Authors:** Nardeen Shafik, Jennifer E. Akpo, Kristie C. Waterfield, William A. Mase

**Affiliations:** Department of Health Policy and Community Health, Jiann-Ping Hsu College of Public Health, Georgia Southern University, P.O. Box 8015, Statesboro, GA 30458, USA; ns11489@georgiasouthern.edu (N.S.); ja21661@georgiasouthern.edu (J.E.A.); wmase@georgiasouthern.edu (W.A.M.)

**Keywords:** COVID-19, vaccination, vaccination hesitancy, autoimmune diseases, public health

## Abstract

As COVID-19 vaccination guidelines were issued by Advisory Committee on Immunization Practices (ACIP) and the Centers for Diseases Control and Prevention (CDC) across the US, each state and clinical provider instituted vaccine implementation and education policies and protocols for high-risk populations. However, current research has shown that while people with autoimmune diseases were listed by ACIP and CDC as a COVID-19 high-risk population, the rate of adherence to implementation and education protocols, as well as the prioritization of this sub-population as a high-risk group, varied among the clinicians and vaccinators thus impacting the hesitancy towards the COVID-19 vaccine and a correlation to low vaccination rates. The purpose of this review was to explore factors of COVID-19 vaccination hesitancy in people living with autoimmune diseases in relation to current implementation and education policies and protocols, as well as ethical and contextual factors, while providing possible implications. COVID-19 vaccine hesitancy in people living with autoimmune disease was greater than in the general population, as demonstrated by increased levels of overall mistrust and fear of potential risk and harmful side effects. Evidence has shown that COVID-19 vaccination is safe and effective for patients with autoimmune diseases. Additionally, the benefits of COVID-19 vaccination outweigh its potential risks and adverse effects in this population. However, the non-adherence to policy and protocols, especially community education protocols, by those providing the vaccination have a negative impact on the overall perception of the vaccine and needs to be addressed at local and state levels in order to protect this population. Future research should provide strategies to guide collaborative efforts between government and local agencies in providing tailored vaccination campaigns to this population. In parallel with policy, COVID-19 vaccination intervention implementation and educational protocols should be developed with evidence-based guidelines for public health and clinical professionals that are targeted at this vulnerable high-risk population.

## 1. Introduction

According to the World Health Organization, there have been more than 760 million cases of COVID-19 and more than 6.8 million deaths globally as of March 2023. Despite global and national efforts in response to the pandemic, recent research affirms that current discussions on COVID-19 preparedness and response remain scarce and fragmented. Although the Biden Administration had anticipated the end of the pandemic by 2023, variants continue to spread rapidly, and vaccination rates in populations such as the elderly are still inadequate [1]. Lower vaccination rates and vaccine hesitancy due to misinformation and mistrust [2] exacerbate negative health outcomes in vulnerable populations such as those with autoimmune diseases. While vaccination refusal has become a major global public health concern [3], a recent review of US patients with private health insurance indicated that 83.3% of COVID-19–related deaths occurred among those with at least one pre-existing comorbidity, including autoimmune diseases [4]. Individuals with autoimmune diseases are considered part of a vulnerable group as they are at a higher risk of complications that may lead to death because of COVID-19. As a result, these patients are considered a priority group for COVID-19 vaccination. However, COVID-19 vaccine hesitancy continues to rise within that group [5].

Current research has shown that individuals with autoimmune diseases have been either excluded or under-sampled in recent vaccine clinical trials, which creates a gap in understanding the efficacy and safety of COVID-19 vaccines within this population group [6]. Although data is scarce, the results from previous studies are reassuring in that there are few adverse side effects of COVID-19 vaccination in those with autoimmune diseases [7]. Regarding the few studies that focus on vaccine efficacy and safety in this population, 82 of 280 participants with autoimmune diseases were vaccinated; of those, only 35 had mild effects, and only 1 patient experienced a disease flare-up [8]. Additionally, significant differences in education levels among autoimmune patients impact vaccine literacy and may lead to low vaccination uptake [9]. Gaur and colleagues found in their study of systematic autoimmune rheumatic disease patients that 69% of their study participants with low education levels (incomplete schooling or no education) were more hesitant to be vaccinated in comparison to 39% of the participants with higher education level (completed school or graduated) [8].

Results of a study comparing COVID-19 vaccination hesitancy among participants with cancer, autoimmune diseases, and other comorbid conditions indicated that autoimmune patients exhibited the greatest vaccine hesitancy (19%) versus cancer patients (13.4%) and Chronic lung diseases (17.8%) [4]. In parallel to this article, another recent study examining vaccine hesitancy in patients with autoimmune and autoinflammatory rheumatological diseases indicated that approximately less than half (40%) of their respondents were motivated to become vaccinated due to fear of being infected [3]. This implies that although people with autoimmune diseases may have a general understanding of the importance of vaccination, a gap in vaccination education persists within this community. Thus, the need for evidence-based intervention protocols for health professionals is imperative so that they can develop population-targeted educational strategies regarding COVID-19 vaccine safety as well as health outcome benefits and risks, in an effort to increase vaccine self-efficacy and reduce vaccine hesitancy in those living with autoimmune diseases.

The spread of COVID-19 has amplified the collaborative response between local, state, and federal levels. The federal government (Centers for Diseases Control and Prevention (CDC), in consultation with Advisory Committee on Immunization Practices (ACIP)) provided guidelines that recommended the prioritization of vulnerable communities. Upon the review of state vaccination policies, most states utilized population demographics and CDC/ACIP guidance in order to provide prioritized vaccine protocols for vulnerable high-risk populations. The review found that most states had listed the elderly (ages 65–74) with pre-existing or underlying conditions, including chronic diseases, as the highest priority on the vaccination schedules and thus targeted their educational strategies to this population. Because of an emphasis on the elderly as an important vulnerable high-risk population, many of the non-elderly people with autoimmune diseases fell through the cracks within this tier system [10]. While many local and state COVID-19 vaccination educational protocols regarding vulnerable populations were in place, the focus on the elderly population appears to have taken precedence, and thus many local and state health professionals would not be in adherence to educational protocols for non-elderly vulnerable populations, such as populations of all ages with autoimmune diseases [11]. Because of an increased non-adherence to COVID-19 educational protocols that could have been utilized to debunk false information and promote the importance of vaccines, vaccination misinformation and mistrust led to greater vaccine hesitancy in persons living with autoimmune diseases [12]. Thus, it is imperative to promote policy protocol adherence initiatives that support the prioritization of those with autoimmune diseases.

COVID-19 has negative health outcome impacts on a large proportion of people with immune or autoimmune diseases. Despite this immense negative impact, this vulnerable population has been excluded or under-sampled from COVID-19 vaccination research. While considerations of vaccination safety and efficacy are at the forefront of current vaccine policy deliberations, gaps, including a lack of targeted policy and education protocol adherence, have resulted in vaccination hesitancy and lower vaccination rates in this population. The aims of this review are (1) to identify leading factors affecting vaccination hesitancy among those living in the autoimmune diseases population and (2) to discuss current COVID-19 vaccination policy adherence and ethical considerations regarding vaccination of those living with autoimmune diseases.

## 2. Materials and Methods

A scoping review was conducted using articles from three research databases: PubMed, SCOPUS, and Google Scholar. Regarding methodology, search protocol, and quality reporting, the PRISMA-ScR (Preferred Reporting Items for Systematic Reviews and Meta-Analyses Extension for Scoping Reviews) checklist was used for this review [13]. A comprehensive search was performed within the databases during January 2023.

### 2.1. Search Strategy

The search keywords developed by the authors were relevant to the aims of the research. When necessary, Boolean operators were used to develop the most productive searches within the chosen databases. The searched keywords for the three databases used are outlined in Table 1.

### 2.2. Study Selection

In the initial search a restriction was applied regarding the year in which the articles were published; due to the relevance to COVID-19, a filter was utilized within the databases so that only articles published after the pandemic began would be listed in the search results. All articles that evaluated COVID-19 vaccine hesitancy, rejection, resistance, and factors associated with vaccine hesitancy among individuals with autoimmune diseases met the eligibility criteria. The key search words used in each of the databases include COVID-19, vaccination, hesitancy, and autoimmune diseases.

Studies were eligible for inclusion if they were published from 2020 onward and met the following criteria: (1) studies published in peer-reviewed journals in the English language; (2) observational studies (quantitative, qualitative, or mixed-method) evaluating or reporting primary data on factors impacting COVID-19 vaccine hesitancy in adults 18 years and older with autoimmune diseases; (3) studies that were conducted using participants who were 18 years and older from the United States; and (4) studies that had full texts available. Studies that were excluded were due to the following reasons: (1) systematic or scoping review; (2) editorials, opinion pieces, commentaries, or letters; (3) book chapters; and (4) studies where the full text was not available.

The electronic search results were exported to EndNote software, and duplicates were removed. The title and abstracts of all search results were evaluated by one of the reviewers. After the exclusion of irrelevant articles, the full texts were evaluated for eligibility for inclusion in the review. The full texts were evaluated by the same reviewer and other reviewers, in which they determined that 30% of the articles met the eligibility criteria. Any disagreement about eligibility for inclusion was discussed among the reviewers and resolved.

## 3. Results

Our search initially produced 508 studies from the databases. A total of 303 duplicates were removed, leaving 205 studies left for screening. After the review of abstracts and titles, 185 studies were removed, and 20 articles were assessed for full-text screening; only 10 articles met the eligibility criteria for inclusion in this review. However, one study was excluded from the review because it had insufficient information related to the study’s characteristics. The result of our search strategy is shown in Figure 1.

### 3.1. Study Characteristics

The main characteristics and findings of the studies included in the review are summarized in Table 2. The sample size of all included studies (nine) was between 101 to 21,943 participants. Seven of the studies were exclusively conducted in the United States. However, two of the studies included participants from the United States as well as other countries. Uhr et al. included participants that lived in countries located on all continents except Antarctica, with most participants (87.2%) living in the United States [14]. The other study by Tsai et al. included participants that lived in multiple countries, including the United States (74.2%), Canada (8.5%), the UK (8.1%), Australia (3.1%), and countries in Europe, Central, South America, the Caribbean, the Middle East, the Russian Federation, Africa, or the Far East (6.1%) [4]. Tsai et al. also included vaccine hesitancy in participants with cancer (27.3%), autoimmune diseases (23.2%), and chronic lung diseases (35.4%) [4]. Smith et al. included participants with other diseases; 54% with respiratory diseases, 61% with an autoimmune disorder, and 57% with autoimmune and respiratory diseases. We included these studies in order to gain additional information regarding vaccine hesitancy in people with autoimmune diseases that may additionally be living with co-morbidities that would impact their vulnerability to COVID-19.

### 3.2. Quality Appraisal

Table 3 summarizes the quality appraisal results. All studies were assessed by Newcastle-Ottawa Scale (NOS) [15]. The NOS can be used in assessing the quality of non-randomized studies. The study is assessed from three main perspectives, which include the selection of study groups, ascertainment of exposure and outcome, and comparability of groups. A star system is assigned based on the perspectives for each of the included studies and given a score. The NOS was used to assess the quality of studies included in this review and was adapted for this scoping review [15]. Two reviewers assessed the risk of bias and resolved any uncertainty. Two studies received 8 stars, three studies received 7 stars, and four studies received 6 stars. Overall, most studies had an average rating of at least 6 stars.

### 3.3. Factors Associated with COVID-19 Vaccine Hesitancy

The most common reasons for hesitancy among participants were vaccine safety, efficacy, fear of adverse reactions, concern vaccine may interfere with medication making treatment ineffective, fear of worsening symptoms, and mistrust. Additional reasons for vaccine hesitancy that were reported by these studies include apprehension about the newness of the vaccine, needing more information, the fast approval process of the vaccine, and the vaccines causing other diseases.

### 3.4. Vaccine Safety, Efficacy, Concern Vaccine May Interfere with Medication and Fear of Adverse Reaction

Seven of the nine articles reported that participants mentioned safety, efficacy, and fear of adverse reactions as one of the factors associated with vaccine hesitancy. Furthermore, Smith et al. found that participants with autoimmune diseases have a significant association with vaccine hesitancy [16]. They reported fear of adverse reactions as a reason for hesitancy (OR = 0.37; CI = 0.14–0.96). In the study by Uhr et al. on COVID-19 vaccine hesitancy in multiple sclerosis (MS), safety and efficacy concerns (*n* = 244, *n* = 122, respectively) were more common among the black race participants [14]. There were also concerns about adverse reactions and effects on multiple sclerosis symptoms. There was a statistically significant difference with *p* < 0.001 in the vaccination behavior between participants who were vaccine willing compared to those who were hesitant.

The vaccine hesitancy rate was 11.9% in the study by Herman et al., which focused on COVID-19 vaccine hesitancy among patients with inflammatory bowel disease [17]. It was higher in younger black and Hispanic participants, and adverse reaction (74%) was the most common reason for hesitancy. Dunculan and Mancuso reported 17% vaccine hesitancy among participants with rheumatic diseases in their study, with 20% of the participants reporting concern about the short- and long-term adverse effects of the vaccine [18]. Tsai et al. reported 19.4% vaccine hesitancy among participants with autoimmune diseases, and it was higher in younger participants and those with less formal education [4]. Ehde et al. reported 20.3% vaccine hesitancy among participants with MS in their study, with 30.4% having major concerns about the long-term side effects of the vaccine [19]. Hesitancy was higher in non-whites, those with lower education, those without a recent flu shot, and those who had lower trust in the Centers for Disease Control and Prevention (CDC).

The summary from these studies [17,18,19] shows that COVID-19 vaccine hesitancy among people with autoimmune diseases is higher in patients who are younger, identify as black or Hispanic, and have lower educational levels.

**Table 2 vaccines-11-01283-t002:** Study Characteristics of Included Studies.

Reference	Study Design/Data Collection	Sample Size	Respondents by Sex (%)	Age Range (Mean)	Race	Main Findings	Reasons for Hesitancy
Bogart et al., 2021 [20]	Telephone interview	101	Cisgender female—16%, Cisgender male—80, transgender female—3%, gender nonconforming—1%, Gay or bisexual—77%.	Mean age 50.3	X	50% showed hesitancy regarding the COVID-19 vaccine or treatment, and a third of participants said they would not get vaccinated or treated. 97% percent endorsed at least one mistrust belief. About 50% or more major prevalent mistrust belief was a concern of withholding information on the vaccine or lack of honesty by the government.	Mistrust beliefs
Duculan et al., 2022 [18]	Telephone interview	112	Female—83, Male—13, Missing—14	22–87 (50)	Asian—8, Black—10, White—82, Latino—13	77% stated they would receive it, 28% had received the first dose, 6% would not get the vaccine, and 17% were hesitant.	Fear of adverse effects, no reason to be vaccinated, distrust of vaccine information, and fear of worsening rheumatic disease symptoms
Ehde et al., 2021 [19]	Cross-sectional online survey	491	Female—81.3, Male—17.3, non-binary—0.4, Transgender—0.2, other/prefer not to say/no answer—0.8	X	White—90.5, Black—2.5, more than one race—4.1, prefer not to say—1.4, other—0.8, American Indian/Alaska Native—0.4, and Asian—0.2	73.8% planned to get the vaccine, 6% had received one vaccine dose, and 20.3% were hesitant.	Efficacy, long-term effects of the vaccine, the vaccine approval process, wanting more information about the vaccine, and the potential impact of the vaccine on their own health conditions.
Herman et al., 2022 [17]	Electronic survey and telephone interview	210	X	Mean age 46.6	White—76.2%, Black—21%, Asian pacific islander or Native Hawaiian—2.4%. Other—1%	88.10% were already vaccinated or wanted to be as soon as possible. The vaccine hesitancy rate was 11.9% and higher in younger black and Hispanic patients.	Adverse reaction, concern that the vaccine may interfere with medication efficiency, concern that medication may make the vaccine ineffective, and safety of the vaccine.
Shaw et al., 2022 [21]	Online survey/thematic analysis	537	Female—84%, Male-unknown, others-unknown	64% were 65 years or older	White—94%, other races-unknown	93% received or intend to receive at least a dose of the vaccine, 83% had concerns about the vaccine among vaccinated and unvaccinated, 71% had concerns about side effects, and 20% had concerns about the effect of the vaccine on DMARD management and flares.	Concern about side effects, doubts about vaccine effectiveness, mistrust, perception of low risk, and concerns about DMARD management /flares
Smith et al., 2022 [16]	Survey through a prolific survey platform	2535 from the initial survey: 478—autoimmune disorder; 618—respiratory diseases; 136—autoimmune disease and chronic respiratory condition; 1303—no condition (healthy control); 589—other chronic conditions. 55% from initial respondents participated; 54% with respiratory diseases, 61% with autoimmune disorders, and 57% with autoimmune and respiratory diseases	X	X	Non-Hispanic White: Respiratory—70.2, Autoimmune—80.4, both—79.4, None—67.7; Black: Respiratory—7.38, Autoimmune—4.6, both—6.4, None—4.4; Hispanic or Latin0: Respiratory—4.1, Autoimmune—4.6, both—2.9, None—4.9; Asian: Respiratory—9.5, Autoimmune—4.4, both—5.2, None—14; Native American: Respiratory—0.98, Autoimmune—0.2, both—1.5, None—0.5; Two or more: Respiratory—7.9, Autoimmune—5.7, both—6.6, None—6.7.	Participants with autoimmune diseases were the only group to have a significant association with a specific cause for vaccine hesitancy or fear or adverse vaccine reaction.	Adverse reaction for those with an autoimmune disorder
Tsai et al., 2022 [4]	Survey	21,943; 74.2% reside in the US, 8.5% in Canada, 8.1% in the UK, 3.1% in Australia, and 6.1% in Europe, Central, South America, and the Caribbean, the Middle East, the Russian Federation, Africa or the Far East. Cancer—27.3%, autoimmune disease—23.2%, chronic lung disease—35.4%	Female—75.9%, Male-unknown, others-unknown	Mean age 56–65	X	18.6% indicated COVID-19 vaccine hesitancy, 10.3% stated they would not receive the vaccine, 3.5% stated they would probably not receive the vaccine, and 4.8% stated they were not sure they would agree to be vaccinated. 25.8% reported they had received one dose of COVID-19 vaccine. 29.6% of US participants had already undergone vaccination. 19.4% with autoimmune diseases reported vaccine hesitancy compared with 18% of those not treated with autoimmune diseases.	Apprehension regarding the newness of the vaccine concerns about the safety of the vaccine and distrust of the development process
Uhr et al., 2022 [14]	Survey via the online iConquerMS platform	1662 active users, and 789 responded. 15 were excluded due to lack of MS diagnosis, and 73 failed to respond to key questions making 701 analyzed respondents. 87.2% of respondents live in the US, and the remaining live in other countries in North America, Africa, Asia, Europe, Oceania, and South America.	X	20 years and older	Race was categorized as white and others. Whites were 656, and other races were 41.	Younger age, racial minorities, and higher functional disability were independently associated with vaccine hesitancy.	Adverse reactions, safety and efficacy, and effect of vaccine on MS symptoms.
Wu et al., 2022 [22]	Survey	306	Female—77.45%, Male—22.45%	≤24–≥75. The median age was 50 years, and the prevalent age group was 45–54 years.	X	66.24% had received the vaccines or planned to be vaccinated, and 33.99% were unlikely to be vaccinated	Vaccine safety concerns, fast vaccine approval, vaccine efficacy, concern about vaccine causing MS relapse, and concern about the vaccine causing other diseases.

Note: X means that the information was not available in included studies.

**Table 3 vaccines-11-01283-t003:** Quality Appraisal of Included Studies.

Reference	Score	Representativeness of Sample	Sample Size	Non-Respondents	Design and Analysis	Statistical Test	Ascertainment of Exposure	Assessment of Outcome
Bogart et al., 2021 [20]	8	*	*	*	*	*	*	*
Duculan et al., 2022 [18]	7		*	*	*	*	*	*
Ehde et al., 2021 [19]	7		*	*	*	*	*	*
Herman et al., 2022 [17]	6		*		*	*	*	*
Shaw et al., 2022 [21]	6	*	*		*		*	*
Smith et al., 2022 [16]	6	*	*		*	*		*
Tsai et al., 2022 [4]	7	*	*		*	*	*	*
Uhr et al., 2022 [14]	8	*	*	*	*	*	*	*
Wu et al., 2022 [22]	8		*		*	*	*	*

* Signifies information was available in included studies.

### 3.5. Fear of Worsening Symptoms, Mistrust, and Vaccine Causing Other Diseases

Four of the nine articles reported another reason for hesitancy, which was participants’ mistrust of COVID-19 information and the COVID-19 vaccine. Bogart et al. assessed general COVID-19 mistrust and COVID-19 vaccine hesitancy among Black Americans living with HIV [20]. Their study showed a high level of general COVID-19 mistrust and hesitancy among this population. Over 60% of participants endorsed withholding information and lack of honesty by the government as the main reasons for mistrust. Over 50% of participants endorsed COVID-19 vaccine hesitancy and treatment, with approximately one-third stating they would not get a COVID-19 vaccine or treatment. Duculan and Mancuso found that 15% of their study participants were distrustful of the information on the vaccine [18]. Tsai and colleagues reported that 48% of vaccine-hesitant participants also mentioned distrust in the government and its ability to ensure the vaccine was safe [4]. Mistrust was a subtheme in the study by Shaw and colleagues [21]. Some participants did not trust the COVID-19 vaccine, while others were not willing to jeopardize their health by taking an experimental shot.

Uhr [14], Duculan and Mancuso [18], and Wu [22] mentioned fear of worsening symptoms. More than one-third of study participants reported fear of the unknown impact on their rheumatic disease and medication, with hesitancy higher in younger black patients [18]. In addition, there were concerns about the effect of the vaccine on MS symptoms and how it will affect their immunosuppressive disease-modifying therapies (DMTs) [14,23]. Wu and colleagues found that vaccine hesitancy in some participants was due to their concerns that the vaccine may cause other diseases, such as myocarditis and pericarditis [17,22].

### 3.6. Apprehension about the Newness of the Vaccine, Needing More Information, the Fast Approval Process of the Vaccine

Three of the nine studies mentioned either apprehension about the newness of the vaccine, the fast approval process, or needing more information as a contributor to COVID-19 vaccine hesitancy among participants. One study reported that 9.1% of participants had major concerns about the vaccine approval process, and 4.9% wanted additional information among the hesitant group [19]. Participants were also asked if religious beliefs, access to the COVID-19 vaccine, and the cost of the vaccine contributed to vaccine hesitancy. No participants endorsed religious belief as a major reason for hesitancy. Additionally, 7.7% reported cost and access to the vaccine as a major concern for hesitancy. All these factors were not of significant concern to those who were willing or hesitant to receive the vaccine [19]. In another study, 53.1% of vaccine-hesitant participants endorsed concern about the newness of the vaccine, which was the most prevalent apprehension. Several demographic factors were associated with vaccine hesitancy. Hesitancy was higher in younger participants between the ages of 26–35 years. Females were more likely to be hesitant than their male counterparts, and people who had a lower educational degree were more hesitant than those with a college or graduate degree. Participants who had conservative political leanings were more likely to be vaccine hesitant than those with liberal political leanings [4]. The fast vaccine approval process and vaccine ingredients had an odds ratio of OR = 8.91, 95% CI = 4.81–16.51; OR = 6.33, 95% CI = 3.74–10.7, respectively, with both being statistically significant at *p* < 0.001 [22]. The number of participants that were hesitant due to the vaccine’s fast approval process and ingredients among the hesitant group were 88 and 67, respectively. It was also observed in this study that people living in rural locations were 20% more likely to be hesitant than those in urban areas among the hesitant group. Younger age, lower education, and females were also found to be more likely to be hesitant [22]. This is similar to the demographic factors associated with vaccine hesitancy in the other included studies.

## 4. Discussion

Current research indicates hesitancy regarding COVID-19 vaccination among those living with autoimmune diseases, which is concerning given that these vulnerable individuals have experienced an increased proportion of mortality from the pandemic [4]. More research is needed to explore vaccine efficacy and safety in this population. Studies have shown that hesitancy among this population stems from fear of disease flaring after vaccination [20]. Although there is a paucity of long-term safety and efficacy data on COVID-19 vaccination in patients with autoimmune diseases, current evidence strongly suggests that the benefits of vaccination outweigh the risks of adverse effects and disease flares [23]. It is critical for stakeholders and leaders within the community to collaborate in prioritizing the pro-health advocacy efforts with regard to this population’s vaccination misinformation in order to effectively reach and promote the health of those within this population.

### 4.1. Factors Related to Vaccine Hesitancy

One of the main concerns reported by communities regarding COVID-19 vaccination hesitancy is the scarcity of experience and background information concerning the science and technology of COVID-19 vaccination, as well as the possible side effects in those with autoimmune diseases [17]. Furthermore, healthcare professionals also expressed concern regarding the scarcity of experience with the new COVID-19 vaccines among those living with autoimmune diseases, suggesting the importance of communication, especially with the results of the ongoing Phase 3 vaccine studies [17]. This indicates that there are gaps in knowledge and communication among the various health sectors, such as public health and healthcare. If healthcare providers report concerns about limited information and knowledge about COVID-19, how is it possible to decrease COVID-19 hesitancy in the public and specifically in vulnerable and/or marginalized communities? Communication gaps and non-adherence to vaccine educational protocols must be addressed to more efficiently decrease vaccine hesitancy among those living with autoimmune diseases.

In addition, engagement with communities to strategize how to overcome COVID-19 mistrust and encourage COVID-19 vaccination uptake is essential to ensure better outcomes for people with autoimmune diseases. It is evident that society’s distrust of the government has played a significant role in guiding society’s aversion towards accepting vaccinations [24]. The internet and various forms of social media allow for rapid sharing of health information; however, COVID-19 misinformation is an issue that persists. To enhance public trust and reduce vaccine hesitancy, it is important to consider and address those factors that lead to mistrust. 

Potential health risks are also major concerns because of a lack of experience with the current COVID-19 vaccinations in those living with autoimmune diseases. Evidence suggests that establishing clear communication strategies and protocols for the COVID-19 vaccination is challenging. Although it is apparent that vaccinations do not pose a more prominent danger than the infections themselves, healthcare providers and communities are still concerned about the potential risk of worsening autoimmune diseases owing to insufficient data [25]. Currently, there is limited data available on the benefits and risks of COVID-19 vaccines in patients with autoimmune diseases [26]. Another concern is the long-term effects of the vaccines on the progression of certain autoimmune diseases, as well as the overall quality of life among those living with autoimmune diseases. Additionally, there is a gap in the knowledge base regarding health implications after vaccination, especially in those with autoimmune diseases or comorbidities [4]. Therefore, it is imperative to include this population in COVID-19 vaccination research to build rapport, increase knowledge dissemination, and continue to provide transparent information and effective communication in the hope of increasing vaccination uptake.

In order to improve the rate of vaccine uptake, there is a need for an increase in policy and protocol adherence that supports the prioritization of those with autoimmune diseases, such as the utilization of tailored vaccination campaigns and communication efforts, as well as the inclusion of this population in more vaccination-related research.

### 4.2. Policy Action & Future Implications

In January 2021, the Department of Health and Human Services, via the CDC and ACIP, encouraged states to vaccinate those over 65 years of age who were vulnerable to high-risk medical conditions [27]. Policies regarding vaccinating those with underlying conditions were higher than the federal guidance in 20 states. This was because the prevalence of major chronic conditions was 24–41% higher in vulnerable counties [10]. It is apparent that 40% of the states strictly prioritized those with comorbidities (above federal levels) to minimize health inequalities. More importantly, it is worth recognizing that only a few states prioritized some socially vulnerable groups, while it was not mentioned at the federal level. Moreover, indigenous communities have been largely ignored in the pandemic response; however, only four states—Oregon, New Jersey, Utah, and Montana—have prioritized indigenous populations for COVID-19 vaccination [10].

Although the federal government led state policy during the pandemic response, there were some intergovernmental conflicts between state and local governments. Across the board, evidence shows that many state and local sectors are in constant competition [28]. States’ policing powers may create friction with local governments as they interfere with local security or liberty. During the pandemic, states employed ceiling pre-emption, which prohibited local governments from requiring anything other than state law [29]. There is limited evidence on the analysis of COVID-19 vaccination policy action and policy adherence in state versus local governments. However, regarding COVID-19 vaccination, federal and state governments were at the forefront of vaccination, whereas local governments followed. Further research is needed to review the local government’s response to COVID-19 vaccination and prioritization. While each county has its own community dynamics and diversity, it is imperative to determine how local sectors responded and their adherence to the state-provided policies and protocols. Understanding local governmental policies regarding COVID-19 vaccination can lead to better service for vulnerable communities that are in need, decreasing health inequalities and shedding light on future interventions to increase vaccination access and self-efficacy [30].

### 4.3. The Common-Good Approach to Ethical Decision-Making

Although the pandemic is new, research is constantly emerging, and there is confidence in the benefits of COVID-19 vaccines, specifically for those living with other comorbidities or autoimmune diseases. Although vaccination is essential for saving all lives, vulnerable populations such as those living with autoimmune diseases should be prioritized due to the higher risk of complications and morbidity due to COVID-19.

This review examined the factors associated with hesitation in COVID-19 vaccination among people with autoimmune diseases. Given the current evidence on COVID-19 and its impact on vulnerable populations, we applied a common-good approach. The focus of this approach is that society comprises individuals whose own good is inexorably linked to that of the community. Therefore, community members are bound to pursue common values and goals [31].

The common-good approach emphasizes that the social policies, social systems, institutions, and environments on which we depend are beneficial to everyone. This review highlights the importance of achieving the action steps for policy development, and adherence is necessary to overcome communication barriers and other obstacles to achieving the common good. In the case presented in this paper, if culturally competent, adequate, and effective communication is utilized to disseminate the safety and benefits of COVID-19 vaccination in vulnerable populations, including those living with autoimmune diseases and other comorbidities, COVID-19 complications, and mortality due to the pandemic should decrease [32]. This implies that tailored campaigns and initiatives are necessary to minimize the difficulties in obtaining accurate COVID-19 vaccination information in our target population [32]. Applying a common-good approach by building rapport with vulnerable communities can increase the focus on appropriate measures and approaches that address the underlying issues that lead to COVID-19 vaccine misinformation and mistrust. Furthermore, collaboration and leadership engagement among various levels of government is imperative to achieve success [33]. Employing the common-good approach can contribute to assessing the reason behind factors that lead to vaccine hesitancy and encourage policymakers to address these issues, leading to better outcomes [7].

Essentially, a common-good approach would ensure COVID-19 vaccines for vulnerable populations that tend to fall in between the accessibility gaps. It is imperative to have an alignment between policymakers, stakeholders, and healthcare providers to achieve the best methods for overcoming COVID-19 vaccine hesitancy and increasing its uptake in vulnerable communities [32,33]. However, traditional medical ethics include all four ethical principles, while public health ethics tend to focus on beneficence, nonmaleficence, and justice over individual autonomy [34]. When looking for approaches to increasing vaccine uptake in vulnerable and marginalized populations, policymakers and health professionals need to consider which ethical approach is the best for not only their communities but also for the sub-populations within their communities. The use of utilitarianism (maximizing benefit to the greatest number of people) or the principle of nonmaleficence (minimizing harms to individuals; thus, benefits outweigh any potential burdens) may not work for every community or population. Individual autonomy, such as the Rights Approach, which allows people to have the dignity to choose freely and for their choices to be respected, also needs to be considered by policymakers when they are discussing the overall health of their communities [35].

## 5. Conclusions

COVID-19 vaccine hesitancy among people with autoimmune diseases is a public health concern, especially because these vulnerable populations are at an increased risk of mortality from COVID-19. Several factors that contributed to vaccine hesitancy in these groups were identified in this study. Although there is limited research on the safety and efficacy of the COVID-19 vaccine, specifically in people with autoimmune diseases, including its long-term effects, previous studies have shown that vaccination is safe and effective for people living with autoimmune diseases [20]. The scarcity of experience, misinformation, lack of education, and government mistrust are primary causes of vaccine hesitancy.

Policy action adherence is imperative in shifting the focus on this population, leading to the opportunity for multi-sectoral collaborations that fill the gaps that exist to increase the uptake and rate of vaccination in this population. Although present policy protocols prioritize COVID-19 vaccination for the elderly with underlying conditions, there are not adequate policy protocols that prioritize those living with autoimmune disease, regardless of age. Effective evidence-based communication strategies that focus on the benefits and safety associated with the vaccine while addressing issues of fear, mistrust, and misinformation can reduce COVID-19 vaccine hesitancy and increase vaccine self-efficacy, especially in those living with autoimmune diseases.

## 6. Limitations and Future Research

This study had some limitations. Our review focuses specifically on COVID-19 vaccine hesitancy among people with autoimmune diseases. In addition, our review only included studies published in peer-reviewed journals. This could have introduced a risk of publication bias. There is also the risk of volunteer bias from the included studies in our review because participants who volunteered to participate may differ systematically from the general population. Furthermore, our research may have missed essential studies that were non-English and conducted in other countries because it was restricted to studies conducted in English and the United States. This is likely to have increased the risk of selection bias in this study.

Future research that utilizes a systems-thinking approach to include effective and coordinated communication with federal, state, and local jurisdictions needs to be conducted. Policies must aim to ensure that guidance issued at the federal level is relevant and sensitive to state and local needs and applicable to vulnerable and/or marginalized populations. Further studies are needed in looking at COVID-19 vaccines in people with autoimmune diseases and providing strategies to utilize policy to implement tailored communication efforts in this population.

## Figures and Tables

**Figure 1 vaccines-11-01283-f001:**
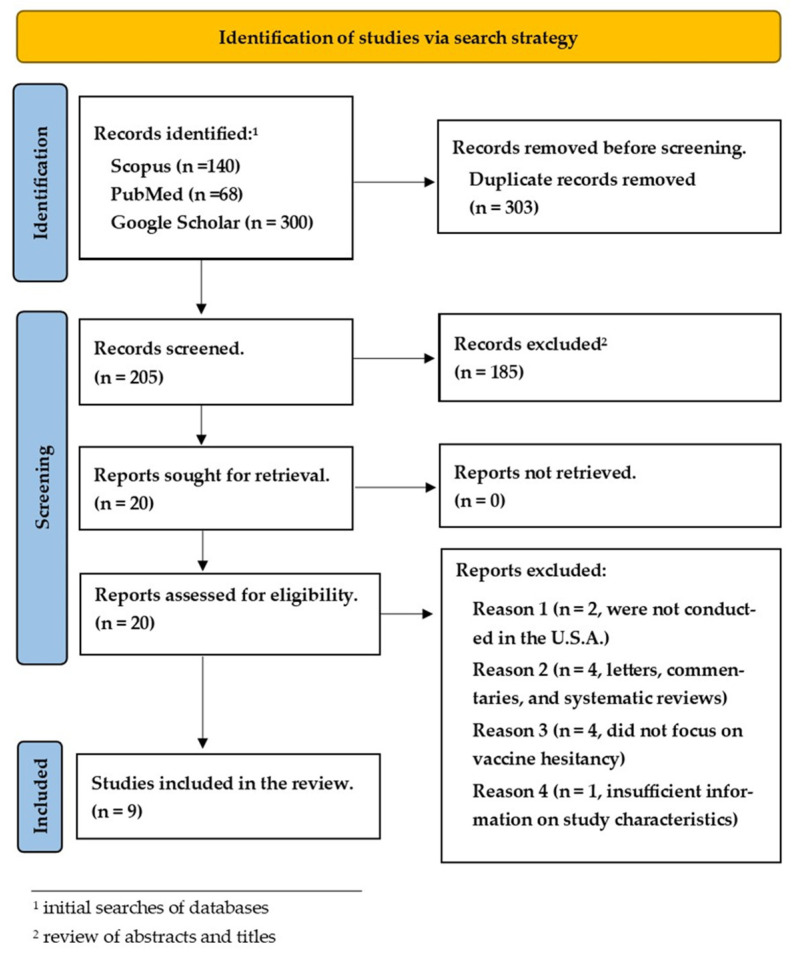
Flow diagram of search and selection according to PRISMA-ScR.

**Table 1 vaccines-11-01283-t001:** Search Strategy and Results for PUBMED, SCOPUS, and Google Scholar.

No.	Search String	Results
Database: PUBMED		
1	covid 19 OR covid-19 OR sars-cov2 OR “novel coronavirus” OR ncov OR 2019ncov OR hcov-19 OR covid19 OR “sarscov 2 infection” OR “severe acute respiratory syndrome coronavirus” OR “wuhan coronavirus”	334,326
2	vaccination OR vaccine OR immunization OR vaccin * OR immune *	4,829,086
3	hesitan* OR anti-vaccin* OR unwillingness	11,182
4	(Autoimmune Diseases[MeSH] OR autoimmun* OR Autoimmunity[MeSH] OR Autoantibodies[MeSH] OR Autoimmune Diseases of the Nervous System[MeSH] OR Neurologic Autoimmun * [tiab] OR Nervous System Immune * [tiab])	684,101
5	# 1 AND # 2 AND #3 AND #4	68
Database: SCOPUS		
6	(“COVID-19” OR “COVID” OR “COVID 19” OR “COVID19” OR “Coronavirus” OR “2019-nCoV” OR “2019 nCoV” OR “2019 Novel Coronavirus” OR “SARS-CoV-2” OR “SARS CoV 2” OR “SARS Coronavirus 2”)) AND (vaccination * OR vaccine * OR immunization * OR vaccin * OR Immun *)) AND (hesitan * OR doubt* OR distrust OR anti-vaccin *)) AND (“Au-toimmune disease” OR “inflammatory bowel disease” OR psoriasis * OR “rheumatic diseases”, OR “systemic lupus erythematosus”))	140
Database: Google Scholar		
7	“Covid 19” OR COVID-19 OR SARS-CoV2 * OR “novel coronavirus” OR ncov* OR “covid19” OR “sarscov 2 infection” OR “severe acute respiratory syndrome” AND vaccination OR immunization AND hesitancy OR “anti-vaccine” AND “Autoim-mune Diseases”	300

Abbreviations: No—Search Number.

## Data Availability

Not applicable.

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
