# Peer review of "COVID-19 Vaccination Hesitancy in Autoimmune Disease Patients: Policy Action and Ethical Considerations"

_vaccines, 2023, doi:10.3390/vaccines11081283_

Round 1

Reviewer 1 Report

This study conducts a systematic literature review to identify common reasons for the COVID-19 vaccine hesitancy among people with autoimmune diseases. Preferred Reporting Items for Systematic Reviews and Meta-Analyses (PRISMA–2009) was used for this review. There are some minor issues need to be addressed:

  1. I kindly suggest that the authors proofread their text. There are some instances of typos throughout the manuscript. For example, * and ** signs stand for or represent something in Figure 1.
  2. The texts on the figure are difficult to read. Also, its sharpness decreases when enlarged. Figures should therefore be in high resolution (TIFF images with a minimum of 300 dpi).
  3. The type of text on the figures is not consistent with the font of the main text.
  4. The search was performed in PubMed, Scopus, and Google scholar databases. Although the review focuses specifically on COVID-19 vaccine hesitancy among people with autoimmune diseases. The number of articles included in the study is so limited (10). WoS database can be included for research and number of articles can be increased.
  5. In conclusion section, the authors propose that adopting a “common good approach” can enhance efforts to address the root causes of COVID-19 vaccine misinformation and mistrust. However, the article acknowledges a significant concern raised by communities regarding vaccine hesitancy, which is the lack of familiarity and understanding of the science and technology behind COVID-19 vaccination. Therefore, to ensure an unbiased perspective, it is crucial to explore alternative solutions and suggestions in the article.
  6. COVID-19 has psychological and socio-economic effects on populations. Please discuss this issue by referring/citing to the following relevant articles “Depression as a mediator between fear of COVID-19 and death anxiety”, “COVID-19 Phobia in the United States: Validation of the COVID-19 Phobia Scale (C19P-SE),” and “Pandemic Awareness Scale (PAS): Evidence of validity and reliability in a Turkish sample during the COVID-19 Pandemic.”
  1. I kindly suggest that the authors proofread their text. There are some instances of typos throughout the manuscript. For example, * and ** signs stand for or represent something in Figure 1.

Author Response

Thank you for your review.  We have attached our responses to your comments. 

Reviewer 2 Report

Authors tried to cover the COVID-19 vaccine hesitancy among population having autoimmune disease for policy making and ethical considerations but I think its not a good study. 

Introduction need extensive revision and authors must need to strengthen the hypothesis why this study is important and what value can be added in literature. 

method need revision, Mesh Details must be added.

I am afraid how authors only found just 10 studies although on google scholar there are alot of studies.

Author must need to recheck the inclusion and redo literature research.

why authors didnot perform meta analysis.

Author Response

(The authors gave the same response as above.)

Reviewer 3 Report

·      The Abstract is inadequate. The research motivation should be explored at the beginning of the Abstract. The main findings should be included as well as the future research direction concisely.

·      Provide more clarification of the objective of the paper so that the reader will understand the main idea of the paper.

·      The motivation of the study needs to be explained better. It is not clear why “autoimmune disorders” were selected as category of interest and what conditions were included. Vaccine clinical recommendations are guided by a risk for VPD and a level of immune suppression. People living with autoimmune disorders can be at different level of risk for both.  

·      The introduction needs to be rewritten. It is long, lacks focus and includes inaccuracies and over generalizations.

·      Elaborate on how this study differs from the existing knowledge on the topic.

·      Methods lack specificity of information. It appears that exclusion criteria were loosely defined. I.e., although patients with autoimmune disorders were the primary goal authors included studies of patients who had other conditions in addition to autoimmune disorders, see page 4: “We included these studies in order to gain additional information re-142 grading vaccine hesitancy in people with autoimmune diseases that may additionally be 143 living with co-morbidities that would impact their vulnerability to COVID-19.” 

·      I am very concerned about only including studies with public access. This sample selection strategy severely limits usefulness and validity of the findings. 

·      Please clarify the utility and validity of New-Castle-Ottawa Scale, its limitations and reasons selected. Please include the interpretation of the scale (i.e., Table 2) 

·      Results:  Discussion section is informative but not adequate in the current form. It needs to revise to be more fully based on the reviewed papers and how it can be applied in future, what is gap in those studies as a future research direction. 
Consider these recent studies on COVID-19 vaccines while revising the discussion. 

·      Add a separate limitation section and future research directions. 

·      Discussion needs to be re-written in its entirety. It is verbose, lacks, focus, includes numerous inaccuracies, including misinformation. Of note, vaccine safety continues to be monitored even after vaccines are authorized and licensed too public. Active vaccine safety surveillance systems allow for access to information about vaccine safety for groups of interest (i.e., Pregnant women, very rare serious adverse events following vaccination, special patient populations)  

The article would benefit from a review by an English speaking reviewer to ensure clarity and accuracy. 

Author Response

(The authors gave the same response as above.)

Reviewer 4 Report

This paper is a scoping review on Covid-19 vaccination in patients with autoimmune disease. The authors include 10 studies to the review, of which one is of low quality. Table 1 does not contain appropriate information on the study group of that particular study (Manansala et al. 2021). The abstract of the review is unconcise. The materials and methods section is short and don't give enough background for the selection of the key words that where used to pick the articles for review. The presentation of the results is inadequate. In summary, the manuscript looks unfinished. Especially the abstract and results require major changes and better presentation. I suggest to drop the paper of Manansala et al. out of the review, if no information on the study population can be presented in Table 1. 

Author Response

(The authors gave the same response as above.)

Round 2

Reviewer 2 Report

Can be accepted 

Author Response

Thank you for your review and comments.  The authors appreciate your feedback. 

Reviewer 3 Report

“The abstract needs to be rewritten. As noted in my initial (first) review and comments. It has not been improved. It has numerous inaccuracies. For example, authors state: “the lack of present policy and prioritization in people with autoimmune diseases has led to expressed hesitancy towards the COVID-19 vaccine and lower vaccination rates among this population. “ or  “The lack of policies that incorporate ethical considerations needs to be addressed 19 to protect this community at local and state levels. Future research should provide strategies to 20 guide collaborative efforts between government and local agencies in providing tailored vaccination 21 campaigns to this population. In parallel with policy, COVID-19 vaccination interventions should 22 be created and targeted towards this vulnerable population, especially with current data.”

People living with autoimmune conditions are included in clinical guidance, which is issued by the ACIP, governing body that provides advice to CDC and clinicians. As such we have guidance for clinicians and vaccinators. Whether clinicians’ access or use it is a separate issue. Authors fail to distinguish between those. Having another policy will not address issues of hesitancy if the information is not accessed and utilized by the providers. I suspect authors are highlighting the importance of public ( and vaccine providers’)  education about safety and benefits of Covid vaccination in this population. But that is a very different message than asking for policy change on how to conduct vaccine clinical trials ( by including people with autoimmune diseases).

Similarly, the introduction along with Discussion are verbose, and includes several inaccuracies. They speak to authors’ lack of understanding of the process CDC has in place when issuing a clinical guidance on Covid vaccination As an example:

“When compar-70 ing federal and state sectors, mMost states prioritized the elderly (ages 65–74) with pre-71 existing or underlying conditions including chronic diseases. Although we recognize that 72 the elderly are an important vulnerable population, non-elderly people with autoimmune 73 diseases fell through the cracks within this tier system [10]. While local sectors followed 74 federal and state recommendations, the focus again was geared specifically towards the 75 elderly, and not necessarily populations of all ages with autoimmune diseases. Because 76 COVID-19 protocols were not in place to debunk false information and promote the im-77 portance of vaccines in this population, vaccination misinformation and mistrust led to 78 great hesitancy [12].Thus, it is imperative to promote develop policy advocacy initiatives 79 to support the prioritization of those with autoimmune diseases. “

People with autoimmune conditions on biologics were highlighted in the clinical guidance documents. NOT JUST ELDERLY. Vaccination recommendations are geared and prioritized by the level of risk.  In fact, they would be included in a high risk group (for vaccination) depending on their level of immune suppression.

The article would benefit from a review by an English-speaking reviewer to reduce redundancy and to enhance clarity. Main points should be succinctly summarized.

Reviewer 4 Report

The authors have revised their manuscript according to the comments of the reviewers. The paper is now acceptable for publication in Vaccines.

Author Response

Thank you for your review and comments.  The authors appreciate all of your feedback.